# SARS-CoV-2 infection is associated with higher chance of diabetes remission among Veterans with incident diabetes

Pandora L. Wander[1,2]*, Elliott Lowy[1], Anna Korpak[1], Lauren A. Beste[1,2], Steven E. Kahn[1,2], Edward J. Boyko[1,2]

1 Veterans Affairs (VA) Puget Sound Health Care System, Seattle, Washington, United States of America
2 Department of Medicine, University of Washington, Seattle, Washington, United States of America

* lwander@u.washington.edu

## Abstract

### Objective

To examine the impact of SARS-CoV-2 on long-term glycemia.

### Research design and methods

We conducted a retrospective inception cohort study using Veterans Health Administration data (March 1, 2020–May 31, 2022) among individuals with ≥1 positive nasal swab for SARS-CoV-2 and individuals with ≥1 laboratory test of any type but no positive swab. Two incident diabetes cohorts were defined based on: 1) a computable phenotype using a combination of diagnosis codes, laboratory tests, and receipt of glucose-lowering medications (n=17,754); and 2) the presence of ≥2 HbA1c results ≥6.5% (n=4,768). We fit log-binomial models examining associations of SARS-CoV-2 with diabetes remission, defined as ≥2 HbA1c measurements <6.5% ≥90 days apart after cessation of any glucose-lowering medications. To help equalize laboratory surveillance of glycemia, we conducted a subgroup analysis among non-hospitalized participants.

### Results

In cohorts 1 and 2 respectively, 25% and 29% had ≥1 positive test for SARS-CoV-2 prior to enrollment, and 21% and 11% had remission. SARS-CoV-2 was associated with a higher chance of remission by both definitions (1: RR 1.22 [95%CI 1.14–1.29]; 2: RR 1.27 [95%CI 1.07–1.50]) over an average 503 (±202) and 494 (±184) days. The association was attenuated among non-hospitalized participants (1: RR 1.11 [1.04–1.20]; 2: R: 1.17 [95%CI 0.97–1.42]).

### Conclusions

Diabetes remission was more common in Veterans with new-onset diabetes after SARS-CoV-2. In non-hospitalized participants, who were likely to have more similar laboratory surveillance, the association was diminished. Differences in surveillance or transient hyperglycemia may explain the observed association.

**Data availability statement:** Data cannot be shared publicly as a condition of approval by the VA Human Research Protection Program at VA Puget Sound. Inquiries may be directed to the VA Puget Sound Research & Development Program at vapugetsoundresearch@va.gov. Data may be available for VA researchers with appropriate approvals.

**Funding:** Funded by a seed grant from the VAPSHCS Office of Research & Development. Salary support was provided to PLW, EL, AK, LAB, SEK and EJB by VAPSHCS and/or the Office of Research & Development. The study sponsor/funder was not involved in the design of the study; the collection, analysis, and interpretation of data; or writing the report; and did not impose any restrictions regarding the publication of the report.

**Competing interests:** The authors have declared that no competing interests exist.

## Introduction

The presence of a prior positive test for SARS-CoV-2/COVID has been associated with an 11% to 276% higher risk of incident diabetes, in particular T2D [1]. To date, existing epidemiological evidence examining the association of SARS-CoV-2 with incident diabetes relies largely on the use of electronic health record (EHR) data. These estimates may be variably subject to bias (e.g., exposure or outcome misclassification, or unequal surveillance). Therefore, the true public health impacts of SARS-CoV-2 on long-term metabolic health and glycemia remain unclear.

Even modest sustained hyperglycemia can lead to the development of complications such as retinopathy, nephropathy, and neuropathy [2]; therefore, the presence of hyperglycemia is an important predictor of the future public health impacts of T2D. Given that a high proportion of humanity has been infected by SARS-CoV-2, the stage is now set for adverse long-term health impacts of this infection that may be identified. Diabetes, and predominantly T2D, may be one such consequence [1]. It is not clear, however, whether this condition remits or becomes chronic, or indeed whether it occurs more frequently at all because care provided for this infection may lead to greater surveillance intensity and potential over-diagnosis compared to the uninfected. T2D remission is characterized by sustained achievement of average blood glucose levels below the diagnostic threshold for diabetes, usually in the absence of glucose-lowering medications, although operational definitions of remission are variable [3]. T2D remission has been reported in secondary analyses of clinical trials of bariatric surgery, intensive lifestyle intervention, and usual diabetes care [4,5]. It can be defined using data from the EHR [6] although detection of remission in the EHR may be impacted by patterns of access to care. Because access to care including receipt of glucose-lowering medications may differ among individuals with and without recent SARS-CoV-2 infection, for this analysis we focused on remission off glucose-lowering medications.

If a true causal relationship exists between SARS-CoV-2 infection and incident diabetes, it may be due to direct β-cell injury, "bystander" effects on β-cells due to infection of other cells in the islet, or systemic effects on insulin resistance or inflammation [7]. Any of these forms of injury, if present, would be expected to impair chances of diabetes remission. Therefore, we hypothesized that the presence of a positive test for SARS-CoV-2 would be associated with a lower chance of EHR-defined remission compared to no positive test among Veterans with incident diabetes. We also hypothesized that receipt of more SARS-CoV-2 vaccination doses might be associated with a higher chance of remission.

## Methods

### Study setting and study population

This analysis was conducted using data from the Corporate Data Warehouse (CDW), a data repository derived from the Veterans Health Administration (VHA) electronic health record. VHA is the largest integrated health care system in the United States [8] and includes a COVID-19 Shared Data Resource which contains analytic variables for all patients tested for SARS-CoV-2 [9].

We identified all patients with ≥ 1 positive nasal swab(s) for SARS-CoV-2 between March 1, 2020, and May 31, 2022. Veterans without a positive nasal swab for SARS-CoV-2 and with a laboratory test performed for any reason between March 1, 2020, and May 31, 2022, were the unexposed comparison group. This comparison group included both individuals with a negative test and individuals with no test at all. The index date was defined as the date the first positive test result was returned in the SARS-CoV-2–negative group and a random date during a month in which they had any laboratory test performed in the SARS-CoV-2–negative group. To allow equal follow-up time between the groups, the distribution of index months among

individuals without a positive test for SARS-CoV-2 was frequency matched to the distribution among SARS-CoV-2–positive individuals. Most SARS-CoV-2 tests were performed in Department of Veterans Affairs (VA) laboratories using US Food and Drug Administration (FDA)–approved RealTime (Abbott Laboratories) or Xpert-Xpress (Cepheid) SARS-CoV-2 assays, but a small number were sent to outside laboratories. Laboratories were required to conform to standards for reporting set out in a VHA operational memo (Feb. 11, 2020) from the Deputy Under Secretary for Health for Operations and Management. Only tests performed by the Public Health Reference Laboratory at the VA Palo Alto Health Care System or by state and local health departments were allowed.

We excluded all individuals with evidence of diabetes prior to their index date, including any laboratory value from plasma or serum (random glucose ≥ 200 mg/dL, fasting glucose ≥ 126 mg/dL, two-hour glucose from an oral glucose tolerance test ≥ 200 mg/dL) or whole blood (HbA1c ≥ 6.5%) [10] or International Classification of Diseases and Related Health Problems (ICD) code consistent with a diagnosis of diabetes. Further, all participants were required to have a laboratory test (serum or plasma glucose or HbA1c) that was not consistent with a diagnosis of diabetes in the 2 years prior to their index date. To avoid capturing Veterans who were not consistently receiving care in VA, we excluded all individuals who did not have an outpatient visit in the year prior to their index month. Because a recent SARS-CoV-2 infection may impact healthcare utilization (e.g., by increasing the frequency of outpatient clinic visits, which may lead to differential detection of previously undiagnosed diabetes by SARS-CoV-2 status), we examined two alternative cohort definitions to identify Veterans with new-onset diabetes.

As we have done previously [11], in cohort 1 we defined incident diabetes using the following computable phenotype: all individuals with incident diabetes after their index date defined by the presence of: (1) two or more abnormal lab values from plasma or serum or whole blood as above [10]; or (2) ICD-10 codes of E08-E13; or (3) receipt of an initial and one refill prescription of a glucose-lowering medication. To avoid capturing transient hyperglycemia related to treatment with systemic corticosteroids, we excluded plasma glucose values collected within 30 days of the index date for all participants. We included diagnostic codes E08 (diabetes mellitus due to underlying condition) and E09 (drug or chemical-induced diabetes mellitus) because we did not want to miss any incident cases of diabetes that were miscoded. We did not include E14 (unspecified diabetes mellitus) because it is not used in the CDW to code diabetes. We also examined associations of SARS-CoV-2 with diabetes remission using a cohort of persons with incident diabetes using a restrictive definition that we believed would be less generalizable but also less likely to be impacted by differences in care-seeking behavior. In cohort 2, we defined incident diabetes as the presence of ≥ 2 HbA1c results ≥ 6.5% after the index date. We defined two different diabetes cohorts with higher sensitivity or higher specificity because no universally agreed upon diabetes phenotype exists based on EHR data. Cohort 1 therefore would be less likely to miss capturing persons with diabetes, while cohort 2 would be less likely to falsely misclassify someone without diabetes as having this condition.

To ensure that the denominator would include only individuals who had the potential to be identified as EHR-defined remission cases, we further restricted the groups to individuals who had ≥ 2 HbA1c results captured in the EHR after diabetes detection. The study was approved by the institutional review board at VA Puget Sound Health Care System (VAPSHCS). The requirement for informed consent was waived for this records review study.

## Variables

We collected data on age; sex; race/ethnicity; body mass index (BMI); tobacco use; Charlson comorbidity index; recent hospitalizations; number of visits, vital sign measurements, plasma/serum glucose results, HbA1c results, and SARS-CoV-2 vaccinations in the year prior to the

index date; and pandemic era of the index date. Veterans specified any number of race/ethnicity responses, which were categorized as selected/not selected for each individual response (e.g., white yes/no). BMI was defined as weight in kg divided by height in meters². Hospitalization was defined as an admission date between 7 days before and 30 days after the index date. Smoking status was classified as current, former, or never based on self-report in VHA's national Health Factors database [12]. If no smoking code was entered, the participant was classified as never smoked. Number of SARS-CoV-2 vaccines received prior to the index date was categorized as 0, 1, or 2 or more. Pandemic eras were defined based on Centers for Disease Control and Prevention genomic surveillance data [13] as "wild type" predominant (March–April 2021), alpha predominant (May–July 2021), delta predominant (August–December 2021), and omicron predominant (January–June 2022).

The remission outcome was defined as ≥2 HbA1c measurements <6.5% 90 or more days apart over all available follow-up time, with the first of these HbA1c measurements occurring ≥90 days after diabetes was first detected. Remission HbA1c results were included only if there were no remaining refills of or dispensed glucose-lowering medications. Remission also required that there were no glucose or HbA1c laboratory values consistent with a diagnosis of diabetes following the HbA1c measurements that qualified for remission.

## Statistical analyses

We examined distributions of covariates according to exposure status in both cohorts. We used all available follow-up time to maximize time to meet the remission definition. We fit log-binomial regression models assessing the association of SARS-CoV-2 with diabetes remission. We chose not to fit Cox regression models for two reasons: 1) because the date of laboratory testing may not correspond to the true date of remission and 2) because the distribution of time to HbA1c testing may differ by the presence of a prior positive test for SARS-CoV-2. We conducted the following sensitivity analyses: examined associations of SARS-CoV-2 with diabetes remission defined as ≥2 HbA1c measurements <5.7% 90 or more days apart; examined associations of SARS-Cov-2 with diabetes remission in non-hospitalized participants only; and examined associations of SARS-CoV-2 with diabetes remission comparing individuals with prior positive vs. a prior negative respiratory swab.

To identify potential confounding variables a priori, we used DAGitty [14] to generate a directed acyclic graph (DAG). We adjusted for variables thought to be causally associated with both likelihood of testing positive for SARS-CoV-2 and with diabetes remission. Analyses were adjusted for age, sex, race/ethnicity, BMI category, Charlson comorbidity index, number of HbA1c results in the year prior to the index date, pandemic era, and number of SARS-CoV-2 vaccinations received. We used multiple imputation with 10 sets of imputations for BMI due to approximately 10% missing values for this variable. Missingness for other variables was negligible. For this analysis, we used α level <0.05 to define statistical significance and did not adjust for multiple comparisons.

## Results

Characteristics of Veterans with newly detected diabetes are shown for both cohorts (Table 1). Respectively, 25%, and 29% had ≥1 positive respiratory swab for SARS-CoV-2. Mean age was 60.7 years old (Cohort 1) and 59.4 years old (Cohort 2), and 21% (n=3,640) and 11% (n=535) met the remission definition. In Cohort 1, compared to Veterans with a positive test for SARS-CoV-2, Veterans without a positive test were older (61.4 years vs 58.7 years, p<0.001), less likely to be hospitalized (4% vs. 21%, p<0.001), had more glucose tests in the year prior to the index date (8.3 vs. 15.5, p<0.001), less likely to be unvaccinated (59% vs. 80%, p<0.001), less

**Table 1. Characteristics of VHA Veterans with newly identified diabetes using two alternative cohort definitions, overall and with stratification by the presence of a positive test for SARS-CoV-2, March 1, 2020–June 1, 2022.**

| | Cohort 1 | | | | Cohort 2 | | | |
|---|---|---|---|---|---|---|---|---|
| | All individuals with new-onset diabetes defined by medications, laboratories, or diagnosis codes | No positive test for SARS-CoV-2 | ≥1 positive test for SARS-CoV-2 | p-value | All individuals with new-onset diabetes defined by ≥2 A1c results ≥6.5% after the index date | No positive test for SARS-CoV-2 | ≥1 positive test for SARS-CoV-2 | p-value |
| n | 17,754 | 13,251 | 4,503 | | 4768 | 3,381 | 1,387 | |
| Follow-up times, days | 503 ±202 | 485 ±194 | 555 ±217 | <0.001 | 503 ±184 | 477 ±180 | 534 ±186 | <0.001 |
| Met remission definition, %yes | 3,640 21% | 2,544 19% | 1,096 24% | <0.001 | 535 11% | 348 10% | 187 13% | 0.002 |
| Age, years | 60.7 ±13.0 | 61.4 ±12.9 | 58.7 ±12.9 | <0.001 | 59.4 ±12.6 | 60.2 ±12.7 | 57.3 ±12.1 | <0.001 |
| Age category, years | | | | <0.001 | | | | <0.001 |
| <40 | 1,185 7% | 807 6% | 378 8% | | 335 7% | 220 7% | 115 8% | |
| 40–49 | 2,462 14% | 1,746 13% | 716 16% | | 738 15% | 487 14% | 251 18% | |
| 50–59 | 4,212 24% | 2,982 23% | 1,230 27% | | 1242 26% | 823 24% | 419 30% | |
| 60–69 | 4,619 26% | 3,522 27% | 1,097 24% | | 1278 27% | 928 27% | 350 25% | |
| 70–79 | 4,407 25% | 3,491 26% | 916 20% | | 993 21% | 771 23% | 222 16% | |
| ≥80 | 869 5% | 703 5% | 166 4% | | 182 4% | 152 4% | 30 2% | |
| Female sex at birth | 1,896 11% | 1,389 10% | 507 11% | 0.145 | 423 9% | 281 8% | 142 10% | 0.034 |
| Race/ethnicity | | | | | | | | |
| Black | 3,909 22% | 2,788 21% | 1,121 25% | <0.001 | 1043 22% | 708 21% | 335 24% | 0.015 |
| Latinx | 1,418 8% | 994 8% | 424 9% | <0.001 | 375 8% | 242 7% | 133 10% | 0.005 |
| White | 12,110 68% | 9,192 69% | 2,918 65% | <0.001 | 3231 68% | 2,321 69% | 910 66% | 0.041 |
| Other | 683 4% | 513 4% | 170 4% | 0.772 | 198 4% | 143 4% | 55 4% | 0.678 |
| BMI, kg/m² | 33.6 ±6.6 | 33.3 ±6.6 | 34.4 ±6.6 | <0.001 | 34.2 ±6.2 | 33.9 ±6.2 | 34.9 ±6.3 | <0.001 |
| BMI category, kg/m² | | | | <0.001 | | | | <0.001 |
| <24 | 1,148 6% | 920 7% | 228 5% | | 182 4% | 135 4% | 47 3% | |
| 24–29 | 3,625 20% | 2,816 21% | 809 18% | | 865 18% | 651 19% | 214 15% | |
| 30–34 | 4,989 28% | 3,694 28% | 1,295 29% | | 1428 30% | 1,005 30% | 423 30% | |
| 35–39 | 3,486 20% | 2,520 19% | 966 21% | | 1059 22% | 740 22% | 319 23% | |
| ≥40 | 2,442 14% | 1,686 13% | 756 17% | | 668 14% | 426 13% | 242 17% | |
| Missing | 2,064 12% | 1,615 12% | 449 10% | | 566 12% | 424 13% | 142 10% | |
| Tobacco use | | | | <0.001 | | | | <0.001 |
| Never | 5,553 31% | 4,304 32% | 1,249 28% | | 1530 32% | 1,144 34% | 386 28% | |
| Former | 6,289 35% | 4,606 35% | 1,683 37% | | 1561 33% | 1,088 32% | 473 34% | |
| Current | 5,912 33% | 4,341 33% | 1,571 35% | | 1677 35% | 1,149 34% | 528 38% | |
| Charlson comorbidity index | 1.0 ±1.6 | 1.0 ±1.6 | 1.0 ±1.6 | 0.934 | 0.8 ±1.4 | 0.8 ±1.4 | 0.8 ±1.5 | 0.980 |
| Hospitalized <7 days before to 30 days after the index date | 1423 8% | 475 4% | 948 21% | <0.001 | 372 8% | 93 3% | 279 20% | <0.001 |
| Number of visits in the year prior to the index date | 19.9 ±21.3 | 18.7 ±19.7 | 23.3 ±24.9 | <0.001 | 17.7 ±19.2 | 16.5 ±17.7 | 20.7 ±22.1 | <0.001 |
| Number of vital signs recorded in the year prior to the index date | 173 ±763 | 123 ±479 | 319 ±1260 | <0.001 | 229 ±994 | 160 ±625 | 398 ±1552 | <0.001 |

(Continued)

**Table 1.** (Continued)

| | Cohort 1 | | | | Cohort 2 | | | |
|---|---|---|---|---|---|---|---|---|
| | All individuals with new-onset diabetes defined by medications, laboratories, or diagnosis codes | No positive test for SARS-CoV-2 | ≥1 positive test for SARS-CoV-2 | p-value | All individuals with new-onset diabetes defined by ≥2 A1c results ≥6.5% after the index date | No positive test for SARS-CoV-2 | ≥1 positive test for SARS-CoV-2 | p-value |
| Number of plasma or serum glucose results in the year prior to the index date | 10.1 ±22.4 | 8.3 ±15.2 | 15.5 ±35.6 | <0.001 | 12.9 ±25.0 | 10.3 ±18.6 | 19.0 ±35.4 | <0.001 |
| Number of A1c results in the year prior to the index date | 0.9 ±0.8 | 0.9 ±0.8 | 0.9 ±0.8 | <0.001 | 0.8 ±0.7 | 0.8 ±0.7 | 0.8 ±0.7 | 0.329 |
| Number of SARS-CoV-2 vaccine doses received prior to the index date | | | | <0.001 | | | | <0.001 |
| 0 | 11,426 64% | 7,807 59% | 3,619 80% | | 3,561 75% | 2,348 69% | 1,213 87% | |
| 1 | 682 4% | 515 4% | 167 4% | | 147 3% | 110 3% | 37 3% | |
| ≥2 | 5,646 32% | 4,929 37% | 717 16% | | 1,060 22% | 923 27% | 137 10% | |
| HbA1c in the 30 days after the index date, % | | | | 0.008 | | | | 0.176 |
| <7.0% | 10,961 62% | 8,171 62% | 2,790 62% | | 2704 59% | 1942 60% | 762 57% | |
| 7.0%–7.9% | 2,151 12% | 1,543 12% | 608 14% | | 1056 23% | 736 23% | 320 24% | |
| 8.0%–8.9% | 550 3% | 402 3% | 148 3% | | 240 5% | 171 5% | 69 5% | |
| ≥9.0% | 1,320 7% | 943 7.1% | 377 8% | | 573 13% | 387 12% | 186 14% | |
| Pandemic era | | | | <0.001 | | | | <0.001 |
| 3/2020–4/2021 (wild type) | 9,683 55% | 6,714 51% | 2,969 66% | | 3,149 66% | 2,108 62% | 1,041 75% | |
| 5/2021–7/2021 (alpha) | 475 3% | 187 1% | 288 6% | | 140 3% | 46 1% | 94 7% | |
| 8/2021–12/2021 (delta) | 6,631 37% | 5,533 42% | 1,098 24% | | 1,315 28% | 1,084 32% | 231 17% | |
| 1/2022–6/2022 (omicron) | 965 5% | 817 6% | 148 3% | | 164 3% | 143 4% | 21 2% | |

Data are presented as mean ± standard deviation, SD for continuous variables and n, % or categorical variables.

Remission was defined as ≥2 A1c measurements <6.5% 90 or more days apart, with the first of these A1c measurements occurring ≥90 days after diabetes was first detected. Remission A1cs were only included if no remaining prescription fills or dispensed medications for glucose-lowering medications. Remission also required that there be no laboratory values consistent with a diagnosis of diabetes between qualifying A1c measurements.

For cohort 1: Incident diabetes was defined by the presence of: (1) two or more abnormal lab values from plasma or serum or whole blood as above [10]; or (2) ICD-10 codes of E08-E13; or (3) receipt of an initial and one refill prescription of a glucose-lowering medication.

For cohort 2: Incident diabetes was defined by the presence of ≥2 HbA1c results ≥6.5% after the index date.

likely to be enrolled during the "wild type" era (51% vs. 66%, p < 0.001), and less likely to meet the remission definition (19% vs. 24%, p < 0.001). In Cohort 2, compared to Veterans with a positive test for SARS-CoV-2, Veterans without a positive test were older (60.2 years vs. 57.3 years, p < 0.001), less likely to have female sex at birth (8% vs. 10%, p = 0.034), less likely to be hospitalized (3% vs. 20%, p < 0.001), had fewer glucose tests in the year prior to the index date (10.3 vs. 19.0, p < 0.001), less likely to be unvaccinated (69% vs. 87%, p < 0.001), less likely to be enrolled during the "wild type" era (62% vs. 75%, p < 0.001), and less likely to meet the remission definition (10% vs. 13%, p = 0.002).

Table 2 lists the adjusted RRs comparing those with or without a positive SARS-CoV-2 test. In Cohort 1, Veterans with a positive test for SARS-CoV-2 were 22% more likely to meet the remission definition compared to Veterans without a positive test (95% CI 1.14–1.29). Other factors associated with a higher chance of remission were all age group older than 60 years compared to age 50–59 years; Black (vs. non-Black) race; BMI < 24 kg/m² and 24–29 kg/m² compared to 30–34 kg/m²; and the presence of more HbA1c results in the year prior to the index

**Table 2. Adjusted RRs (95% CI) for EHR–defined remission * over all follow-up time among Veterans with or without a respiratory swab positive for SARS–CoV–2 using two alternative cohort definitions.**

| | Cohort 1 | | Cohort 2 | |
| --- | --- | --- | --- | --- |
| | n = 17,754 | | n = 4,768 | |
| | RR | 95% CI | RR | 95% CI |
| SARS–CoV–2 positive | 1.22 | 1.14–1.29 | 1.27 | 1.07–1.50 |
| Age category, years (ref = 50–59) | | | | |
| <40 | 0.93 | 0.81–1.08 | 0.67 | 0.44–1.02 |
| 40–49 | 0.90 | 0.80–1.01 | 0.75 | 0.56–1.01 |
| 60–69 | 1.25 | 1.15–1.36 | 1.00 | 0.80–1.26 |
| 70–79 | 1.56 | 1.43–1.70 | 1.37 | 1.09–1.72 |
| ≥80 | 1.80 | 1.59–2.04 | 1.59 | 1.09–2.32 |
| Female sex at birth | 1.08 | 0.98–1.19 | 1.00 | 0.74–1.36 |
| Race/ethnicity | | | | |
| Black | 1.15 | 1.02–1.30 | 1.18 | 0.84–1.66 |
| Latinx | 1.07 | 0.95–1.20 | 1.08 | 0.78–1.50 |
| White | 1.00 | 0.89–1.12 | 0.98 | 0.72–1.34 |
| Other | 0.95 | 0.79–1.14 | 1.39 | 0.91–2.12 |
| BMI category, kg/m² (ref = 30–34) | | | | |
| <24 | 1.24 | 1.12–1.38 | 1.18 | 0.83–1.68 |
| 24–29 | 1.14 | 1.06–1.23 | 1.14 | 0.92–1.41 |
| 35–39 | 0.91 | 0.83–0.99 | 0.82 | 0.65–1.04 |
| ≥40 | 0.84 | 0.75–0.93 | 0.86 | 0.66–1.12 |
| Number of A1c results in the year prior to the index date | 1.16 | 1.12–1.20 | 1.29 | 1.17–1.43 |
| Number of SARS–CoV–2 vaccine doses received prior to the index date (ref = 0) | | | | |
| 1 | 0.95 | 0.81–1.12 | 1.02 | 0.63–1.63 |
| ≥2 | 0.71 | 0.63–0.80 | 0.58 | 0.41–0.82 |
| Pandemic era (ref = 3/2020–4/2021 [wild type]) | | | | |
| 5/2021–7/2021 (alpha) | 0.82 | 0.68–1.00 | 0.95 | 0.58–1.55 |
| 8/2021–12/2021 (delta) | 0.77 | 0.70–0.86 | 0.90 | 0.68–1.21 |
| 1/2022–6/2022 (omicron) | 0.61 | 0.50–0.74 | 0.68 | 0.34–1.34 |

* Defined as < 6.5% 90 or more days apart over all available follow-up time, with the first of these HbA1c measurements occurring ≥ 90 days after diabetes was first detected.

Models were additionally adjusted for region (Midwest, Southeast, South, West, and Other).

date. Factors associated with a lower chance of remission were age younger than 50 years; BMI ≥35 kg/m²; receipt of ≥2 vaccine doses; and delta or omicron-era enrollment. In Cohort 2, Veterans with a positive test for SARS-CoV-2 were 27% more likely to meet the remission definition compared to Veterans without a positive test (95%CI 1.07–1.50). Other factors associated with a higher chance of remission were age of 70 years and above and more HbA1c results in the year prior to the index date. Factors associated with a lower chance of remission were age <40 years and receipt of ≥2 vaccine doses. Results were similar in a sensitivity analysis in which we defined remission as ≥2 HbA1c measurements <5.7% 90 or more days apart (Table 3).

Table 4 presents differences in results among non-hospitalized participants. In Cohort 1, a positive test for SARS-CoV-2 was associated with a 11% higher chance of remission compared to no positive test (95%CI 1.04–1.20). Other factors associated with a higher chance of remission among non-hospitalized individuals were age 60 years and above compared to age 50–59 years; BMI below 30 kg/m² compared to 30–34 kg/m²; and more HbA1c results in the

**Table 3. Adjusted RRs (95% CI) for EHR–defined remission (defined as ≥2 HbA1c <5.7% over 90 days or more)\* over all follow-up time among Veterans with or without a respiratory swab positive for SARS–CoV–2 using two alternative cohort definitions.**

| | Cohort 1 | | Cohort 2 | |
|---|---|---|---|---|
| | n = 17,754 | | n = 4,768 | |
| | RR | 95% CI | RR | 95% CI |
| SARS–CoV–2 positive | 1.16 | 1.08–1.25 | 1.20 | 1.01–1.44 |
| Age category, years (ref = 50–59) | | | | |
| <40 | 0.66 | 0.55–0.80 | 0.46 | 0.27–0.76 |
| 40–49 | 0.82 | 0.72–0.93 | 0.71 | 0.52–0.96 |
| 60–69 | 1.22 | 1.11–1.35 | 0.96 | 0.76–1.20 |
| 70–79 | 1.58 | 1.43–1.74 | 1.36 | 1.08–1.72 |
| ≥80 | 1.74 | 1.50–2.02 | 1.58 | 1.08–2.33 |
| Female sex at birth | 1.02 | 0.91–1.15 | 1.00 | 0.73–1.37 |
| Race/ethnicity | | | | |
| Black | 1.26 | 1.09–1.45 | 1.24 | 0.87–1.77 |
| Latinx | 1.10 | 0.96–1.25 | 1.15 | 0.82–1.60 |
| White | 0.98 | 0.86–1.12 | 0.98 | 0.71–1.34 |
| Other | 1.06 | 0.86–1.30 | 1.34 | 0.85–2.09 |
| BMI category, kg/m² (ref = 30–34) | | | | |
| <24 | 1.02 | 0.90–1.17 | 1.14 | 0.79–1.65 |
| 24–29 | 1.12 | 1.03–1.23 | 1.15 | 0.93–1.44 |
| 35–39 | 0.91 | 0.82–1.00 | 0.83 | 0.65–1.06 |
| ≥40 | 0.82 | 0.72–0.93 | 0.85 | 0.64–1.13 |
| Number of A1c results in the year prior to the index date | 1.16 | 1.11–1.21 | 1.29 | 1.17–1.44 |
| Number of SARS–CoV–2 vaccine doses received prior to the index date (ref = 0) | | | | |
| 1 | 0.91 | 0.75–1.11 | 0.96 | 0.57–1.59 |
| ≥2 | 0.67 | 0.59–0.77 | 0.57 | 0.40–0.83 |
| Pandemic era (ref = 3/2020–4/2021 [wild type]) | | | | |
| 5/2021–7/2021 (alpha) | 0.88 | 0.71–1.10 | 0.98 | 0.59–1.63 |
| 8/2021–12/2021 (delta) | 0.78 | 0.70–0.88 | 0.90 | 0.67–1.22 |
| 1/2022–6/2022 (omicron) | 0.67 | 0.54–0.83 | 0.73 | 0.37–1.45 |

\* Defined as <5.7% 90 or more days apart over all available follow-up time, with the first of these HbA1c measurements occurring ≥90 days after diabetes was first detected.

Models were additionally adjusted for region (Midwest, Southeast, South, West, and Other).

year prior to the index date. Factors associated with a lower chance of remission were age < 40 years and receipt of ≥ 2 vaccine doses. Among non-hospitalized participants in Cohort 2, a positive test for SARS-CoV-2 was not associated with a higher chance of remission. Factors associated with a higher chance of remission were age 70 years and greater and more HbA1c results in the year prior to the index date. Factors associated with a lower chance of remission were age below 50 years and receipt of ≥ 2 vaccine doses. Characteristics of Veterans with a prior positive vs. prior negative nasal swab for SARS-CoV-2 and sensitivity analyses conducted in this group are shown in **Supplementary Tables 1 and 2**. Results were similar to the overall cohort. Results were also similar in a sensitivity analysis that additionally adjusted for HbA1c in the 30 days after the index date (data not shown).

## Discussion

In this large national cohort of Veterans with incident diabetes, diabetes remission was common. The presence of a positive respiratory swab for SARS-CoV-2 was associated with a

**Table 4. Adjusted RRs (95% CI) for EHR-defined remission (defined as ≥ 2 HbA1c < 6.5% over 90 days or more) over all follow-up time among non-hospitalized Veterans with or without a respiratory swab positive for SARS-CoV-2 using two alternative cohort definitions.**

| | Cohort 1 | | Cohort 2 | |
| --- | --- | --- | --- | --- |
| | n = 16,331 | | n = 4,396 | |
| | RR | 95% CI | RR | 95% CI |
| SARS–CoV–2 positive | 1.11 | 1.04–1.20 | 1.17 | 0.97–1.42 |
| Age category, years (ref = 50–59) | | | | |
| <40 | 0.95 | 0.81–1.11 | 0.67 | 0.43–1.05 |
| 40–49 | 0.89 | 0.79–1.01 | 0.80 | 0.59–1.09 |
| 60–69 | 1.21 | 1.10–1.33 | 1.04 | 0.81–1.32 |
| 70–79 | 1.56 | 1.42–1.71 | 1.45 | 1.14–1.85 |
| ≥80 | 1.79 | 1.56–2.05 | 1.69 | 1.13–2.52 |
| Female sex at birth | 1.09 | 0.98–1.21 | 1.04 | 0.75–1.42 |
| Race/ethnicity | | | | |
| Black | 1.13 | 0.99–1.30 | 1.24 | 0.86–1.78 |
| Latinx | 1.08 | 0.96–1.23 | 1.12 | 0.79–1.58 |
| White | 0.98 | 0.86–1.11 | 0.98 | 0.71–1.36 |
| Other | 0.91 | 0.74–1.11 | 1.22 | 0.77–1.94 |
| BMI category, kg/m$^2$ (ref = 30–34) | | | | |
| <24 | 1.22 | 1.08–1.37 | 1.19 | 0.82–1.74 |
| 24–29 | 1.15 | 1.06–1.25 | 1.15 | 0.92–1.45 |
| 35–39 | 0.91 | 0.83–1.01 | 0.84 | 0.66–1.08 |
| ≥40 | 0.84 | 0.74–0.94 | 0.89 | 0.67–1.18 |
| Number of A1c results in the year prior to the index date | 1.17 | 1.13–1.22 | 1.32 | 1.18–1.46 |
| Number of SARS–CoV–2 vaccine doses received prior to the index date (ref = 0) | | | | |
| 1 | 0.98 | 0.82–1.17 | 1.13 | 0.70–1.83 |
| ≥2 | 0.74 | 0.66–0.84 | 0.61 | 0.42–0.88 |
| Pandemic era (ref = 3/2020–4/2021 [wild type]) | | | | |
| 5/2021–7/2021 (alpha) | 0.79 | 0.63–0.99 | 1.04 | 0.61–1.76 |
| 8/2021–12/2021 (delta) | 0.73 | 0.66–0.82 | 0.87 | 0.64–1.20 |
| 1/2022–6/2022 (omicron) | 0.59 | 0.48–0.73 | 0.56 | 0.26–1.22 |

Models were additionally adjusted for region (Midwest, Southeast, South, West, and Other).

higher chance of diabetes remission among individuals with diabetes of recent onset compared to no prior positive test over an average of roughly 500 days of follow-up. The association was attenuated among non-hospitalized participants, suggesting that differences in clinical behavior by SARS-CoV-2 status (and not biological effects from the virus) might account for the differences seen. Increased surveillance after SARS-CoV-2 may contribute to a higher likelihood of detecting remission in this population. Alternatively, more intensive diabetes management among persons with recent SARS-CoV-2 might contribute to the higher rates of remission in this group. To our knowledge, this study is the first to report associations of SARS-CoV-2 infection with diabetes remission among individuals with diabetes of recent onset. Point estimates for nearly all associations were generally similar in cohorts 1 and 2.

Limited comparisons can be made between the previous studies of remission (which were done in the pre-COVID era) and the current analysis. Remission was common in our cohort compared to previous studies examining diabetes remission in non-surgical populations (21% in Cohort 1 and 11% in Cohort 2). In a secondary analysis of the Action for Health in Diabetes (Look AHEAD) trial, the incidence of any remission (partial or complete) at 1 year was 11.5% (95%CI 10.1–12.8%) in the intensive lifestyle intervention arm and 2.0% (95%CI 1.4–2.6%) in the diabetes support and education arm [5]. In the Diabetes and Aging Study, an EHR T2D cohort derived from Kaiser Northern California data during the pre-COVID era, 1.6% (95% 1.5%–1.7%) had remission over 7 years [6]. However, the cumulative incidence of remission in individuals with diabetes thought to be present for < 2 years was 4.55% (95%CI 4.25–4.88), more similar to our estimate but still lower. In an administrative claims study using Optum Labs data, the proportion of individuals meeting three alternative remission definitions ranged from 0.8 to 2.3% [15]. In a cross-sectional population-based study of Scottish adults, 4.8% of individuals with prior diagnosis of diabetes met a remission definition [16]. Older age was associated with higher chance of remission in several analyses [6,16], including ours, but other factors, such as higher BMI, were not consistently associated with remission. Several aspects of study design might contribute to the differences seen. First, there were large differences in duration of diabetes at the time that remission was assessed. To our knowledge, this study is the first to examine remission in an inception diabetes cohort. Thus, the high rates of remission we found may be due in part to glycemic variability near the diagnostic thresholds for diabetes around the time of diabetes diagnosis. Second, inclusion and exclusion criteria differed across the studies. For example, in Look AHEAD, participants had overweight/obesity and underwent glycemic testing at baseline and prospectively at protocol-defined future times. Finally, we had much a much lower proportion of female participants. We are not aware of previous analyses examining sex/ gender differences in diabetes remission. This is an important area for future research.

We initially hypothesized that SARS-CoV-2 infection might lead to beta-cell injury or sustained insulin resistance and therefore a lower chance of diabetes remission. However, we found the opposite. Reasons that SARS-CoV-2 might be associated with a higher chance of remission are not clear. One possibility is that COVID may cause a transient perturbation in glucose metabolism that recovers with time and is different from the typical onset of T2D due to progressive insulin resistance and a gradually failing beta cell. Some of this could be related to the impact of illness on counterregulatory hormones that can affect insulin sensitivity, the β cell and thus glycemia [17]. Another possibility is that the observed association may reflect differences in care-seeking patterns, in particular increased surveillance among individuals with a recent SARS-CoV-2–related illness. The association of SARS-CoV-2 infection with higher chance of remission was attenuated in the sensitivity analysis restricted to non- hospitalized individuals. In this non-hospitalized subgroup, surveillance is expected to be more similar between SARS-CoV-2–positive and–negative groups, which provides some support for this explanation. Differences in surveillance might impact not only EHR-defined

diabetes remission but also EHR-defined diabetes detection, as we have previously reported [11]. Existing literature on the association of COVID and incident diabetes relies on proxies for diabetes diagnosis (diagnosis codes, laboratory tests collected for clinical care, and glucose-lowering medication prescriptions) which may incorrectly classify people as having incident diabetes when in truth diabetes is not present. In the setting of increased surveillance around the time of COVID, this form of misclassification might be more common in persons with recent COVID. Together, these findings might argue that impacts of SARS-CoV-2 on metabolic health may be more modest that early estimates have suggested.

Surprisingly, we found that receipt of ≥2 SARS-CoV-2 vaccination doses was associated with a lower chance of remission, also counter to the direction that we had hypothesized. This observation may also reflect differences in care-seeking behavior. For example, persons who receive recommended vaccines may also be more likely to receive recommended diabetes screening, testing, and treatment (both before and during the pandemic). Thus, they may be better represented in the data in terms of diagnosis codes, laboratory results, and medication usage. If this phenomenon is present, the vaccinated group might appear to be less likely to have remission when in reality, this association is due to differences in selection (i.e., the vaccinated group with EHR-defined diabetes has comparatively more "true" diabetes than unvaccinated or less vaccinated groups). The potential impact of these and other complex patterns in EHR data should prompt caution in readers tempted to infer the presence of causal (biological) effects of COVID on diabetes or its remission solely based on estimates derived from EHR data.

Our study has several strengths, including a large national sample of racially diverse adults. We also took steps to address important sources of bias that may impact inference in EHR studies: misclassification bias and surveillance bias. We also used two different strategies to define incident diabetes. The first definition integrated diagnosis codes, laboratory values, and receipt of glucose-lowering medications, which may be more sensitive for detection of incident diabetes than strategies that rely entirely on diagnostic codes [18]. The second definition required the presence of ≥2 HbA1c results ≥6.5%. This definition is more restrictive and expected to be more specific for incident diabetes. To reduce misclassification of prevalent diabetes, we used diagnosis codes, laboratory results, and glucose-lowering medication use. We also required that all participants had a laboratory test (serum or plasma glucose or hemoglobin HbA1c) that was not consistent with a diagnosis of diabetes in the 2 years prior to their index date. To address surveillance bias, we restricted the analysis to individuals with at least one visit in the year prior to enrollment to capture a population that was engaged in VA-provided health care. Because hospitalization may lead to increased healthcare utilization, and the proportion hospitalized was much greater in the SARS-CoV-2–positive group, we conducted a secondary analysis restricted to non-hospitalized individuals.

There are also some important limitations. First, findings may not be generalizable to populations that differ demographically from patients receiving care from the VA. Generalizability may be further reduced by the fact that the analytic cohorts were restricted to individuals with several HbA1c values available in the EHR; however, characteristics were similar across the cohorts (Table 1), suggesting the findings may be broadly applied in similar populations. Diagnosis of diabetes was based on data available in the EHR and did not include, e.g., clinical symptoms; however, we did use established criteria based on glycemia measurements as per American Diabetes Association recommendations [10], which do not require clinical symptoms to establish the diagnosis. We did not identify SARS-CoV-2 infections that occurred in the absence of VA care, nor did we identify SARS-CoV-2 infections after the index date. In general, these forms of misclassification would be expected to bias findings toward the null. Finally, this analysis did not examine impacts of pandemic-related weight loss or gain, which may contribute to these associations.

## Conclusions

A United States Veterans with incident diabetes, EHR-defined remission was common. A positive test for SARS-CoV-2 was associated with higher chance of remission. Mechanisms underlying this observed association remain unclear but may reflect greater surveillance among individuals with recent SARS-CoV-2. Studies examining the long-term impacts of SARS-CoV-2 infection on glycemia and metabolic health are needed.

## Supporting information

**S1 and S2 Tables. Characteristics of Veterans with a prior positive vs. prior negative nasal swab for SARS-CoV-2 and sensitivity analyses conducted in this group.**
(XLSX)

## Author contributions

**Conceptualization:** Pandora L. Wander, Edward J. Boyko.

**Formal analysis:** Elliott Lowy.

**Funding acquisition:** Pandora L. Wander.

**Writing – original draft:** Pandora L. Wander.

**Writing – review & editing:** Elliott Lowy, Anna Korpak, Lauren A. Beste, Steven E. Kahn, Edward J. Boyko.

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
