## [Decision Letter · Decision Letter 0]

19 Sep 2024

PONE-D-24-04293SARS-CoV-2 infection is associated with higher chance of diabetes remission among Veterans with incident diabetesPLOS ONE

Dear Dr. Wander,

Thank you for submitting your manuscript to PLOS ONE. After careful consideration, we feel that it has merit but does not fully meet PLOS ONE’s publication criteria as it currently stands. Therefore, we invite you to submit a revised version of the manuscript that addresses the points raised during the review process.

The manuscript has been evaluated by three reviewers, and their comments are available below.

The reviewers have raised a number of major concerns. They request improvements to the reporting of methodological aspects of the study, as well as some clarity in the results reported.

Could you please carefully revise the manuscript to address all comments raised?

We look forward to receiving your revised manuscript.

Kind regards,

Avanti Dey, PhD

Staff Editor

PLOS ONE

Journal Requirements:

Additional Editor Comments (if provided):

Reviewers' comments:

Reviewer's Responses to Questions

**Comments to the Author**

1. Is the manuscript technically sound, and do the data support the conclusions?

Reviewer #1: Yes

Reviewer #2: Partly

Reviewer #3: Yes

2. Has the statistical analysis been performed appropriately and rigorously? 

Reviewer #1: Yes

Reviewer #2: Yes

Reviewer #3: Yes

3. Have the authors made all data underlying the findings in their manuscript fully available?

Reviewer #1: Yes

Reviewer #2: No

Reviewer #3: Yes

4. Is the manuscript presented in an intelligible fashion and written in standard English?

Reviewer #1: Yes

Reviewer #2: Yes

Reviewer #3: Yes

5. Review Comments to the Author

Reviewer #1: The manuscript describes appropriate methodology, controls, and analyses. The conclusions drawn are based on the data available but they take this as not sufficient evidence to conclude that there might be some surveillance bias in it. The statistical analysis has been performed rigorously, data availability is adequate, and the manuscript is presented in an intelligible manner with standard English.

A potential bias in this study could be the presence of stress-induced hyperglycemia or misdiagnosis of diabetes mellitus in the included patients. The author should provide additional details regarding the diagnostic criteria for diabetes in the patients included in this study, which are whether it only relied on the registered International Classification of Diseases (ICD) or if it also considered the patients' clinical symptoms.

The author could add novelty about how patients who experienced remission were more likely to have received receipt of ≥2 vaccine doses.

Reviewer #2: Thank you, authors, for sharing your work on the association between incident diabetes remission and SARS-COV-2.

I have a few queries concerning the methods and results

1. Why was it important to have 2 separate diabetes cohorts? Were the authors unsure about the validity of one of the diagnostic mechanisms? Could you present the combined results?

2. The SARS-CoV-2–negative group included individuals who were not tested; why were they included? Is the assumption that there was no asymptomatic COVID or that all symptomatic would have been tested? Could we have an analysis excluding this group?

3. From Table 1 it seems that persons with SARS-CoV-2 were followed up for approximately 2 months longer that persons without SARS-CoV-2. Could that be an explanation for the difference in outcomes? Could the authors standardise the follow-up time or report a ‘time to outcome’ analysis?

4. Could the authors include the average initial glycaemic index in each group? And if it associated with remission rates?

5. Table 3 defined remission as ≥2 HbA1c measurements <5.7% 90 or more days apart. According to the Methods section, this was a criteria for inclusion of data into the study. Can the authors then please explain what was the data in Table 2?

Minor comments

Abstract:

The Objective is not in the correct grammatical format

Methods

“To avoid capturing transient hyperglycemia related to treatment with systemic corticosteroids, we excluded plasma glucose values collected within 30 days of the index date for all participants” Does this include persons with SARS-COV-2 not treated with steroids? Persons -ve for SARS-COV-2?

In the next line, the authors stated that drug or chemical-induced diabetes mellitus was included. Doesn’t this include steroid-induced diabetes?

Conclusion

“suggesting that increased surveillance after SARS-CoV-2 may contribute to a higher likelihood” – stricter management may also contribute.

Reviewer #3: 1. The objective is not clearly stated, it is a research question.

2. There is no Discussion section, instead everything is included in conclusions. It would be methodologically more appropriate to include both sections or the conclusions at the end of the discussion.

6. PLOS authors have the option to publish the peer review history of their article (what does this mean? ). If published, this will include your full peer review and any attached files.

**Do you want your identity to be public for this peer review?** For information about this choice, including consent withdrawal, please see our Privacy Policy .

Reviewer #1: **Yes: ** NUR ROCHMAH

Reviewer #2: No

Reviewer #3: **Yes: ** Frank Hernández-García

---

## [Author Response · Author response to Decision Letter 1]

30 Oct 2024

Reviewers' comments:

Reviewer #1: The manuscript describes appropriate methodology, controls, and analyses. The conclusions drawn are based on the data available but they take this as not sufficient evidence to conclude that there might be some surveillance bias in it. The statistical analysis has been performed rigorously, data availability is adequate, and the manuscript is presented in an intelligible manner with standard English.A potential bias in this study could be the presence of stress-induced hyperglycemia or misdiagnosis of diabetes mellitus in the included patients.

1. The author should provide additional details regarding the diagnostic criteria for diabetes in the patients included in this study, which are whether it only relied on the registered International Classification of Diseases (ICD) or if it also considered the patients' clinical symptoms.

We thank the reviewer for this comment. Information about the patients’ clinical symptoms was not available. We added the following text to the Discussion (page 13, line 5): “Diagnosis of diabetes was based on data available in the EHR and did not include, e.g., clinical symptoms; however, we did use established criteria based on glycemia measurements as per American Diabetes Association recommendations, which do not require clinical symptoms to establish the diagnosis.”

2. The author could add novelty about how patients who experienced remission were more likely to have received receipt of ≥2 vaccine doses.

We added the following text to the Discussion (page 12, line 1): “Surprisingly, we found that receipt of ≥2 SARS-CoV-2 vaccination doses was associated with a lower chance of remission, also counter to the direction that we had hypothesized. This observation may also reflect differences in care-seeking behavior. For example, persons who receive recommended vaccines may also be more likely to receive recommended diabetes screening, testing, and treatment (both before and during the pandemic). Thus, they may be better represented in the data in terms of diagnosis codes, laboratory results, and medication usage. If this phenomenon is present, the vaccinated group might appear to be less likely to have remission when in reality, this association is due to differences in selection (i.e., the vaccinated group with EHR-defined diabetes has comparatively more “true” diabetes than unvaccinated or less vaccinated groups). The potential impact of these and other complex patterns in EHR data should prompt caution in readers tempted to infer the presence of causal (biological) effects of COVID on diabetes or its remission solely based on estimates derived from EHR data.”

Reviewer #2: Thank you, authors, for sharing your work on the association between incident diabetes remission and SARS-COV-2. I have a few queries concerning the methods and results

1. Why was it important to have 2 separate diabetes cohorts? Were the authors unsure about the validity of one of the diagnostic mechanisms? Could you present the combined results?

We thank the reviewer for this question. We used two different cohort definitions because we believed that a recent SARS-CoV-2 infection might impact healthcare utilization. For example, a recent SARS-CoV-2 infection might lead to more outpatient clinic visits, which could lead to differential patterns of laboratory testing and diagnosis coding. In this way, detection of previously undiagnosed diabetes would differ by SARS-CoV-2 status) due to differences in opportunity for diagnosis. We added the following text to the Methods to further explain our use of two cohorts (page 6, line 18): “We also examined associations of SARS-CoV-2 with diabetes remission using a cohort of persons with incident diabetes using a restrictive definition that we believed would be less generalizable but also less likely to be impacted by differences in care-seeking behavior.”

2. The SARS-CoV-2–negative group included individuals who were not tested; why were they included? Is the assumption that there was no asymptomatic COVID or that all symptomatic would have been tested? Could we have an analysis excluding this group?

We now include an analysis excluding individuals who were not tested for SARS-CoV-2. These results are similar to the findings in the main paper and are presented in Supplementary Tables 1 and 2.

3. From Table 1 it seems that persons with SARS-CoV-2 were followed up for approximately 2 months longer that persons without SARS-CoV-2. Could that be an explanation for the difference in outcomes? Could the authors standardise the follow-up time or report a ‘time to outcome’ analysis?

We appreciate this concern. For the primary analysis, we chose not to fit a Cox regression model for two reasons: 1) because the date of laboratory testing may not correspond to the true date of remission and 2) because the distribution of time to HbA1c testing may differ by the presence of a prior positive test for SARS-CoV-2. We conducted a sensitivity analysis in which we standardized the follow-up time by excluding random participants without a positive test for SARS-CoV-2 until the distribution of follow-up days was similar between the groups. Using this cohort, we again fit log-binomial regression models to assess the association of SARS-CoV-2 with diabetes remission. Results were very similar to the overall cohort (Table R1, below).

4. Could the authors include the average initial glycaemic index in each group? And if it associated with remission rates?

We presume that the reviewer would like to see a measure of average glycemia, such as initial HbA1c at the time that diabetes was identified. We have included this information in Table 1. As the reviewer has suggested, we examined the association of initial HbA1c with remission. We also conducted a sensitivity analysis including initial HbA1c in the multivariate model testing the association of SARS-CoV-2 with remission. Results are shown in Table R2 below. We added the following text to the Results (page 9, line 26): “Results were also similar in a sensitivity analysis that additionally adjusted for HbA1c in the 30 days after the index date (data not shown).”

5. Table 3 defined remission as ≥2 HbA1c measurements <5.7% 90 or more days apart. According to the Methods section, this was a criteria for inclusion of data into the study. Can the authors then please explain what was the data in Table 2?

We apologize for the confusion on this point. Persons were excluded from the cohort if they had any laboratory value from plasma or serum (random glucose ≥200 mg/dL, fasting glucose ≥126 mg/dL, two-hour glucose from an oral glucose tolerance test ≥200 mg/dL) or whole blood (HbA1c ≥6.5%) (10) or International Classification of Diseases and Related Health Problems (ICD) code consistent with a diagnosis of diabetes prior to their enrollment in the cohort. To meet the remission definition in this sensitivity analysis, participants had to * have a HbA1c <5.7% 90 or more days apart over all available follow-up time, with the first of these HbA1c measurements occurring ≥90 days after diabetes was first detected. We added the following footnote to Table 3: “* defined as <5.7% 90 or more days apart over all available follow-up time, with the first of these HbA1c measurements occurring ≥90 days after diabetes was first detected.”

Minor comments:

Abstract: The Objective is not in the correct grammatical format

We have revised the abstract Objective: “To examine the impacts of SARS-CoV-2 on long-term glycemia.”

Methods: “To avoid capturing transient hyperglycemia related to treatment with systemic corticosteroids, we excluded plasma glucose values collected within 30 days of the index date for all participants” Does this include persons with SARS-COV-2 not treated with steroids? Persons -ve for SARS-COV-2? In the next line, the authors stated that drug or chemical-induced diabetes mellitus was included. Doesn’t this include steroid-induced diabetes?

The reviewer is correct. No glucose values collected within 30 days of the index date were used to meet criteria for incident diabetes for any participant regardless of SARS-CoV-2 status or corticosteroid use. These early glucose values were not considered because we wanted to avoid misclassifying transient steroid-induced hyperglycemia (as a treatment for COVID) as new-onset diabetes.

The reviewer is also correct that E09 (drug- or chemical-induced diabetes) may be used for some individuals who have steroid-induced diabetes. For the purposes of this analysis, we wanted to capture all cases of new onset diabetes regardless of whether steroid exposure might be in the causal pathway of hyperglycemia. E09 was the sole ICD code reported for only 33 individuals in the dataset; thus, impacts of the inclusion of this code are anticipated to be modest.

Conclusion: “suggesting that increased surveillance after SARS-CoV-2 may contribute to a higher likelihood” – stricter management may also contribute.

Thanks for this suggestion. We have revised the Discussion (page 10, line 5) as follows to acknowledge this possibility: “The association was attenuated among non-hospitalized participants, suggesting that differences in clinical behavior by SARS-CoV-2 status (and not biological effects from the virus) might account for the differences seen. Increased surveillance after SARS-CoV-2 may contribute to a higher likelihood of detecting remission in this population. Alternatively, more intensive diabetes management among persons with recent SARS-CoV-2 might contribute to the higher rates of remission in this group.”

Reviewer #3:

1. The objective is not clearly stated, it is a research question.

We have revised the abstract to state a clear objective: “To examine the impacts of SARS-CoV-2 on long-term glycemia.”

2. There is no Discussion section, instead everything is included in conclusions. It would be methodologically more appropriate to include both sections or the conclusions at the end of the discussion.

As suggested, we have revised the manuscript to include both a Discussion section and a Conclusions section.

---

## [Decision Letter · Decision Letter 1]

16 Dec 2024

PONE-D-24-04293R1SARS-CoV-2 infection is associated with higher chance of diabetes remission among Veterans with incident diabetesPLOS ONE

Dear Dr. Wander,

Thank you for submitting your manuscript to PLOS ONE. After careful consideration, we feel that it has merit but does not fully meet PLOS ONE’s publication criteria as it currently stands. Therefore, we invite you to submit a revised version of the manuscript that addresses the points raised during the review process.

We look forward to receiving your revised manuscript.

Kind regards,

Anna Denee’ Ware, MPH, MS

Academic Editor

PLOS ONE

Journal Requirements:

Reviewers' comments:

Reviewer's Responses to Questions

**Comments to the Author**

1. If the authors have adequately addressed your comments raised in a previous round of review and you feel that this manuscript is now acceptable for publication, you may indicate that here to bypass the “Comments to the Author” section, enter your conflict of interest statement in the “Confidential to Editor” section, and submit your "Accept" recommendation.

Reviewer #2: (No Response)

2. Is the manuscript technically sound, and do the data support the conclusions?

Reviewer #2: Yes

3. Has the statistical analysis been performed appropriately and rigorously? 

Reviewer #2: Yes

4. Have the authors made all data underlying the findings in their manuscript fully available?

Reviewer #2: (No Response)

5. Is the manuscript presented in an intelligible fashion and written in standard English?

Reviewer #2: Yes

6. Review Comments to the Author

Reviewer #2: Thanks for the opportunity to review again.

1. In the previous round of reviews, I asked about the importance of having 2 separate diabetes cohorts. The answer given was that the authors believed that a recent SARS-CoV-2 infection might impact healthcare utilization and thereby diagnosis rates. Whereas I understand and agree with this view, I am not sure it answers the question posed. There are 2 diabetes cohorts, one defined by lab values and prescriptions and the other cohort defined by A1C. Each cohort is further stratified by COVID status. The comparison I am querying is between those in cohort 1 who have T2DM and COVID and those in cohort 2 who have T2DM and COVID. Why were these data separated? Why were there separate definitions for T2DM?

2. New comment: In the discussion, a section was added on remission by vaccine status. Introductory blurb in the introduction or objective sections would be useful.

7. PLOS authors have the option to publish the peer review history of their article (what does this mean? ). If published, this will include your full peer review and any attached files.

**Do you want your identity to be public for this peer review?** For information about this choice, including consent withdrawal, please see our Privacy Policy .

Reviewer #2: No

---

## [Author Response · Author response to Decision Letter 2]

23 Dec 2024

19 December 2024

Dear editors:

Thank you for your thoughtful review and for the opportunity to revise this manuscript. We respond to the reviewers’ comments point-by-point below.

Great gratitude,

Luke

Pandora Lucrezia “Luke” Wander, MD, MS, FACP

Associate Professor of Medicine

Adjunct Associate Professor of Epidemiology & Public Health Genetics

University of Washington

Staff Physician

Veterans Affairs Puget Sound Health Care System

Reviewers' comments:

1. In the previous round of reviews, I asked about the importance of having 2 separate diabetes cohorts. The answer given was that the authors believed that a recent SARS-CoV-2 infection might impact healthcare utilization and thereby diagnosis rates. Whereas I understand and agree with this view, I am not sure it answers the question posed. There are 2 diabetes cohorts, one defined by lab values and prescriptions and the other cohort defined by A1C. Each cohort is further stratified by COVID status. The comparison I am querying is between those in cohort 1 who have T2DM and COVID and those in cohort 2 who have T2DM and COVID. Why were these data separated? Why were there separate definitions for T2DM?

Thank you for the opportunity to clarify. As there is no generally accepted ideal EHR-based phenotype for new-onset diabetes, we chose two methods, a “specific” method that would yield a higher positive predictive value and minimize false positive diagnoses, and a “sensitive” method that would be more likely to capture true positive diagnosis but also be more likely to include a higher number of false positive diagnoses. A specific EHR definition such as the one we used for cohort 2, which required two HbA1c values ≥6.5% for inclusion, runs a low risk of including individuals who do not truly have diabetes, but may result in the exclusion of so many individuals in the dataset that generalizability is low. On the other hand, a sensitive EHR definition such as the one we used for cohort 1, which used all available data on ICD10 codes, glucose values, HbA1c values and glucose-lowering medication use, is more generalizable but may result in inclusion of individuals who do not truly have diabetes. To balance these priorities, we chose to examine the association of SARS-CoV-2 with remission in both cohorts. Importantly, all individuals included in the specific (cohort 2) were also included in the more sensitive cohort (cohort 1). To be explicit, the population with COVID and T2D in cohort 2 is a subset of the population with COVID and T2D in cohort 1. Our results demonstrate in general similar point estimates for effect measures, thereby increasing confidence in their validity across broader and more targeted diabetes definitions. We added the following text to the methods (p. 6, line 23): “We defined two different diabetes cohorts with higher sensitivity or higher specificity because no universally agreed upon diabetes phenotype exists based on EHR data. Cohort 1 therefore would be less likely to miss capturing persons with diabetes, while cohort 2 would be less likely to falsely misclassify someone without diabetes as having this condition.”

2. New comment: In the discussion, a section was added on remission by vaccine status. Introductory blurb in the introduction or objective sections would be useful.

We added the following text to the introduction (p. 5, line 2): “We also hypothesized that receipt of more SARS-CoV-2 vaccination doses might be associated with a higher chance of remission.”

---

## [Editor Report · Decision Letter 2]

27 Dec 2024

SARS-CoV-2 infection is associated with higher chance of diabetes remission among Veterans with incident diabetes

PONE-D-24-04293R2

Dear Dr. Wander,

We’re pleased to inform you that your manuscript has been judged scientifically suitable for publication and will be formally accepted for publication once it meets all outstanding technical requirements.

Kind regards,

Anna Denee’ Ware, MPH, MS

Academic Editor

PLOS ONE
---

## [Editor Report · Acceptance letter]

PONE-D-24-04293R2

PLOS ONE

Dear Dr. Wander,

I'm pleased to inform you that your manuscript has been deemed suitable for publication in PLOS ONE. Congratulations! Your manuscript is now being handed over to our production team.

Kind regards,

on behalf of

Dr. Anna D. Ware

Academic Editor

PLOS ONE